# The Dark Side of Sphingolipids: Searching for Potential Cardiovascular Biomarkers

**DOI:** 10.3390/biom13010168

**Published:** 2023-01-13

**Authors:** Paola Di Pietro, Carmine Izzo, Angela Carmelita Abate, Paola Iesu, Maria Rosaria Rusciano, Eleonora Venturini, Valeria Visco, Eduardo Sommella, Michele Ciccarelli, Albino Carrizzo, Carmine Vecchione

**Affiliations:** 1Department of Medicine, Surgery and Dentistry “Scuola Medica Salernitana”, University of Salerno, 84081 Baronissi, Italy; 2Vascular Physiopathology Unit, IRCCS Neuromed, 86077 Pozzilli, Italy; 3Department of Pharmacy, University of Salerno, 84084 Fisciano, Italy

**Keywords:** sphingolipids, cardiovascular diseases, cerebrovascular diseases, ceramides, sphingosine-1-phosphate

## Abstract

Cardiovascular diseases (CVDs) are the leading cause of death and illness in Europe and worldwide, responsible for a staggering 47% of deaths in Europe. Over the past few years, there has been increasing evidence pointing to bioactive sphingolipids as drivers of CVDs. Among them, most studies place emphasis on the cardiovascular effect of ceramides and sphingosine-1-phosphate (S1P), reporting correlation between their aberrant expression and CVD risk factors. In experimental in vivo models, pharmacological inhibition of de novo ceramide synthesis averts the development of diabetes, atherosclerosis, hypertension and heart failure. In humans, levels of circulating sphingolipids have been suggested as prognostic indicators for a broad spectrum of diseases. This article provides a comprehensive review of sphingolipids’ contribution to cardiovascular, cerebrovascular and metabolic diseases, focusing on the latest experimental and clinical findings. Cumulatively, these studies indicate that monitoring sphingolipid level alterations could allow for better assessment of cardiovascular disease progression and/or severity, and also suggest them as a potential target for future therapeutic intervention. Some approaches may include the down-regulation of specific sphingolipid species levels in the circulation, by inhibiting critical enzymes that catalyze ceramide metabolism, such as ceramidases, sphingomyelinases and sphingosine kinases. Therefore, manipulation of the sphingolipid pathway may be a promising strategy for the treatment of cardio- and cerebrovascular diseases.

## 1. Introduction

Sphingolipids, including sphingosine, ceramide, sphingosine-1-phosphate (S1P) and ceramide-1-phosphate, are bioactive components in cell membranes that participate in and regulate numerous biological processes, such as cell proliferation and survival, maturation, senescence and apoptosis [1,2,3]. Sphingolipids can be synthesized via the de novo synthesis pathway, but they can also be formed through the sphingomyelinase pathway and/or the so-called “salvage” pathway (Figure 1).

Over the last decade, growing numbers of studies have highlighted the role of sphingolipids in the pathogenesis of CVDs [4,5,6,7,8,9]. In mice and rats, repression of sphingolipid biosynthesis attenuates cardiometabolic risk factors, including glucose intolerance, insulin resistance, diabetes, hypertension, atherosclerotic plaque development, arterial dysfunction and heart failure (HF) [10,11,12,13,14]. Data from patients have also indicated associations of tissue and circulating levels of sphingolipids with increased risk of CVDs, including HF, hypertension, metabolic syndrome and coronary artery disease (CAD) [12,15,16,17,18].

The aim of this article is to summarize recent clinical and experimental findings suggesting sphingolipids as potential cardiovascular risk biomarkers and drug targets.

## 2. Atherosclerosis and Coronary Artery Disease

Sphingolipids, and in particular, ceramide, can contribute to the pathogenesis of atherosclerosis [19], an inflammatory and potentially lethal condition characterized by the generation of atheromatous plaques, consisting of cholesterol and other lipids [20], in medium- and large-sized arteries [21]. Ceramide can be generated from sphingomyelin (SM) via the activation of the de novo synthesis pathway or from sphingomyelinases (SMases). The development of an atherosclerotic lesion involves a large number of proinflammatory cytokines, such as tumor necrosis factor α (TNF-α) and interleukin-1β (IL-1β) [22], which, along with oxidized low-density lipoprotein (oxLDL), can stimulate ceramide generation via sphingomyelin hydrolysis [23]. In their work, Laulederkind and colleagues [24] demonstrated that C2-ceramid is able to induce interleukin 6 (IL-6) gene expression in human fibroblasts, a cytokine known to be involved in inflammation [25] and for its ability to induce the liver’s production of the greatest predictor of future cardiovascular risk that has direct proinflammatory effects: C-reactive protein [26]. It has also been proven that oxLDL stimulates an enzymatic cascade that includes neutral SMase, ceramidase and sphingosine kinase, thereby promoting the production of S1P to stimulate mitogenesis [27] and proliferation of smooth muscle cells (SMCs) [28,29], a hallmark of atherosclerotic lesion development. Moreover, Li and collaborators [30] showed that endogenous ceramides contribute to the subendothelial infiltration of oxLDL into the vessel wall.

Deficiency in or pharmacologic inhibition of neutral SMase2 in an Apolipoprotein E (ApoE)-null mouse model resulted in a decrease in atherosclerotic lesions and macrophage infiltration and lipid deposition via a mechanism involving the nuclear factor erythroid 2-related factor 2 (Nrf2) pathway [31]. S1P is also released from activated platelets and interacts with endothelial cells during the atherosclerotic process [32]. It has also been proven that S1P can induce platelet shape change and aggregation [33] and that ceramide stimulates the release of the plasminogen activator inhibitor (PAI-1) [23,34,35], contributing, once again, to a variety of pathophysiological processes, such as thrombosis and atherosclerosis. In line, high endogenous S1P levels exaggerated atherosclerotic lesion development and increased plasma cholesterol levels in ApoE-null mice [36].

As mentioned above, tumor necrosis factor is able to activate a neutral sphingomyelinase [37], causing an increase in ceramide that, during the inflammation process, can act as an intermediate in TNF signaling in endothelial cells [38]. Furthermore, treatment of these cells with a water-soluble synthetic C8-ceramide or sphingomyelinase C led to the evidence that ceramide can induce NF-κB translocation to the nucleus and an increase in surface expression of E-selectin [38], which mediates the interaction of leukocytes with endothelium [39], thus, contributing to the initial stage of the disease. These sphingolipids can also evoke endothelial cell apoptosis [40,41,42], causing plaque erosion and unleashing other complications combined with the atherosclerotic process [23].

Another mechanism involved in the plaque formation induced by sphingolipids is represented by ceramide’s inclination to self-aggregate. It has a primary role in atherosclerosis’s onset since it contributes to the accumulation of LDL rich in this sphingolipid [43,44]. This notion is also supported by the evidence that ceramide content in LDL present in the atheromatous plaques is significantly higher than the plasma LDL and this sphingolipid was found only in aggregated LDL [45]. Furthermore, ceramide and other sphingolipids’ levels were increased in human atherosclerotic plaques and associated with plaque inflammation and instability [46,47].

Several clinical trials have highlighted the role of sphingolipids in atherosclerosis, reporting increased plasma concentrations of ceramides, sphingomyelins, sphinganine and sphingosine in patients with CAD [48]. An interesting clinical trial through a large-scale metabolomic analysis on a total of 200 patients tried to identify potential biomarkers for early-stage atherosclerosis. Analyses showed increased levels of 24 metabolites and a decrease in another 18 metabolites. Overall, nine metabolites were found to be suitable as combinatorial biomarkers, with an acceptable diagnostic accuracy [49]. In patients with CAD, the new PCSK9 has also been shown to significantly alter plasma lipidome composition and not only lipoprotein particles (LDL-C). Although the target for CVD prevention remains the lowering of LDL-C, the lipidome represents an asset for cardiovascular risk prediction and modification [50]. Similarly, treatment with fenofibrate showed not only the expected decrease in triglycerides, LDL-C and total cholesterol, but also an independent reduction in ceramide levels and of plasma apoC-II, apoC-III, apoB100 and SMase, with an increase in apoA-II and adiponectin levels [51]. A bi-ethnic angiographic case-control study showed increased plasma levels of SM in patients with coronary artery disease [48]. This was observed in both African-American and white participants, with a multivariate logistic regression analysis independent of other cardiovascular risk factors [48]. Using an unbiased machine learning approach, Poss et al. [52] identified 30 sphingolipids that were significantly elevated in the serum of patients with CAD (n = 462) compared with healthy controls (n = 212). Circulating ceramides were strongly correlated with disease severity since their levels were higher in subjects with CAD severity and major adverse cardia and cerebrovascular events (MACEs) [52,53,54,55]. Moreover, elevated levels of specific ceramide species (C16:0, C18:0, C22:0, C24:0 and C24:1) were associated with increased thrombotic risk, adverse CAD incidents and all-cause mortality [8,56,57,58,59,60]. Based on these findings, the authors of the study suggest serum ceramides as powerful biomarkers of CAD that could be useful to improve risk stratification [52].

## 3. Heart Failure

One of the major mechanisms connecting ceramides to impaired cardiomyocyte function can be ascribed to their pathological actions in mitochondria, where they can accumulate, increase permeability to cytochrome c and ultimately initiate apoptosis [12,61,62]. Accumulation of ceramide has been reported to drive insulin resistance, oxidative stress and mitochondrial dysfunction in human-induced pluripotent stem-cell-derived cardiomyocytes [63]. Experimental myocardial infarction (MI) performed on C57B/L6 mice induced altered sphingolipid metabolism, with increased expression of the first and rate-limiting enzyme in the de novo pathway, serine palmitoyl transferase (SPT), and consequently increased ceramide levels [12]. In the same study, failing myocardium had elevated levels of the serine-palmitoyl transferase long-chain 2 subunit (SPTLC2), and its overexpression resulted in a marked accumulation of cellular ceramide, together with increased apoptosis in a human cardiomyocyte cell line. These observations suggest a role for the de novo ceramide synthesis pathway. Accordingly, inhibition of SPT with myriocin decreased C16:0, C24:1 and C24:0 ceramide accumulation, prevented adverse cardiac remodeling [12] and reduced infarcted area, oxidative stress and inflammatory markers [64]. In line with these in vivo findings, lipidomic analysis revealed significantly increased ceramide levels in the myocardium and serum of patients with advanced HF [12], and its circulating levels were associated with adverse cardiac outcome during a median follow-up of 4.7 years [16]. Recently, sphingolipid metabolism gene dysregulation was found in HF human cardiac tissue, with the major changes occurring in the expression of genes involved in the de novo and salvage pathways [65]. Even more interesting, the authors demonstrated, for the first time, that, along with ceramide elevation, S1P is enhanced in HF cardiac tissue [65].

Only one clinical trial has associated plasma ceramides and sphingomyelins with heart failure. In this study, plasma Cer and SM species were analyzed with Cox regression to highlight the risk of incident HF. In a large cohort of 4249 older adults, higher plasma levels of ceramide and sphingomyelin were indicative of increased heart-failure risk, in-dependently of sex, age, race, body mass index (BMI) and baseline coronary heart disease, whereas higher levels of C22:0, SM20:0, SM22:0 and SM24:0 were indicative of a lower risk of heart failure [66].

## 4. Hypertension

Hypertension is a major cause of death and disability in Europe and in the rest of the World, responsible for an estimated 8 million deaths and 148 million disability life-years lost worldwide in 2015 [67].

Although some clinical and experimental studies reported altered sphingolipid metabolism in hypertension [7,68,69], the mechanism through which S1P promotes disease onset and propagation remains mainly elusive. Previous work conducted by Spijkers et al. [7] showed increased total ceramide and sphingosine circulating levels in a spontaneously hypertensive rat (SHR) model. Moreover, the authors observed, for the first time, an increase in ceramide level in patients with stage 1–3 hypertension and its concentration correlated with disease severity [7].

Deficiency in the rate-limiting enzymes in the generation of S1P, sphingosine kinases (SphK) 1 and 2, resulted in a significant decrease in blood pressure as well as artery contractility in angiotensin II (AngII)-induced hypertension in the wild type (C57BL/6J) [68,70]. Accordingly, S1P plasma levels were positively correlated with systolic BP in in a murine model of AngII-mediated slowly developing hypertension [71]. Yogi and co-workers [69] reported a novel molecular process that, starting from S1P, induced vascular inflammation in stroke-prone SHR rats (SHRSP) through epidermal growth factor receptor (EGFR) and platelet-derived growth factor (PDGFR) transactivation. The authors demonstrated that these effects were abrogated by the use of VPC23019, a potent S1P receptor (S1PR) 1 inhibitor, thus, indicating the role of the S1P1 receptor in this process. 

Recently, our group elucidated the molecular mechanism by which dysregulated sphingolipid metabolism is involved in hypertension [72]. We demonstrated that in vivo administration of recombinant sortilin—a member of the vacuolar protein sorting 10 protein family of sorting receptors—altered the sphingolipid balance by initiating a signaling cascade that, starting from acid SMase activation, leads to increased levels of S1P at the expense of ceramides [72]. Mechanistically, sortilin acts as a pathological modulator of bioactive sphingolipids, through which it impairs endothelial function and leads to high blood-pressure levels, effects that were mediated by an oxidative-stress-dependent activation mechanism [72]. 

We and others [71,72] have shown that circulating levels of S1P are elevated in humans with arterial hypertension. Through proteomic profiling, Jujic and collaborators [71] revealed that plasma S1P was associated with increased systolic blood pressure, multiple cardiovascular, inflammation and metabolism biomarkers. Importantly, these findings were observed in a relatively young study cohort with very few cardiovascular incidents, thus, indicating that these alterations might be associated with the pathogenesis of CVDs rather than the end stage of the disease [71].

In line with this elegant study and in a translational approach, we identified S1P as a powerful biomarker associated with high blood pressure, since the increase in circulating S1P (alongside soluble NADPH oxidase 2-derived (NOX2-derived) peptide) was more pronounced in uncontrolled hypertensive patients and linearly correlated with each other in the entire hypertensive cohort [72]. The only clinical study available was conducted on 920 patients in Beijing between 2016 and 2018 with a mean follow-up of 2.3 years [73]. These hypertensive patients’ plasma was evaluated using ultra-performance liquid chromatography–tandem spectrometry and the risk of MACE, including acute coronary syndrome, heart failure, stroke and CV death, was correlated with sphingolipid levels. The plasma levels of the 71 patients that experienced MACE showed a significant increase in three specific ceramides (d18:1/16:0, d18:1/22:0 and d18:1/24:0), all of which were highly significant in predicting MACE. This clinical trial highlights the possible use of sphingolipids as a biomarker for improving the identification of hypertensive patients at high risk of CVD [73].

## 5. Stroke

Through an in vivo experimental model of stroke, Kim and co-workers [74] demonstrated the critical role of S1PR2 in the induction of cerebrovascular permeability, neurovascular injury and intracerebral hemorrhage. In particular, pharmacological or genetic deficiency in S1PR2 resulted in a marked decrease in both infarct ratio and total cerebral oedema ratio and improved neurological scores after transient focal ischemia induced by middle cerebral artery occlusion (MCAO) [74]. Moreover, global alterations in sphingolipid metabolism were observed in the brain tissue of two spontaneously hypertensive models, the stroke-prone (SHRSP) and the stroke-resistant (SHRSR) rat strains [75].

The activity of the rate-limiting kinase in the generation of S1P, SphK1, has also been reported to be detrimental in ischemia-induced brain injury [76]. Pharmacological inhibition or siRNA-mediated knockdown of Sphk1 reduced infarct volumes and improved the neurological deficits after MCAO via the attenuation of pro-inflammatory mediators in the cortical penumbra [76]. Long-chain ceramides were markedly increased in mouse brain tissue after 1 h of MCAO [77]. In vivo experiments performed in mice revealed a dramatic increase in circulating sphingolipid levels 24 h after the injury in the stroke compared to sham animals [78]. In particular, the authors identified C42:1 and SM36:0 as the top-performing species, with an increase of up to 60-fold. These findings were further corroborated in a small cohort of patients with acute ischemic stroke, in which the two sphingolipid species identified in the animal model positively correlated with the severity of injury [78]. Recently, plasma ceramide levels were found to be significantly increased in patients with acute ischemic stroke with large artery occlusion and cerebral small-vessel disease [79,80,81]. Even more interestingly, higher levels of Cer (d18:1/18:0), Cer (d18:1/20:0) and Cer (d18:1/22:0) significantly correlated with poor functional outcomes 3 months after stroke [79]. A line of clinical research born towards the end of the 1980s has investigated the use of gangliosides in stroke. Gangliosides are compounds belonging to the general class of glycolipids, particularly abundant in the brain. They owe their name to the fact that they were isolated for the first time in the ganglia and are functionally qualified constituents of membrane receptor sites to which specific effectors bind, for example, neurotransmitters, hormones, bacterial toxins, etc., to evoke specific responses at the synapse level. Starting from 1984, with a double-blind clinical trial with monosialoganglioside (GM1) [82], various attempts have been made to identify possible beneficial effects from these molecules in stroke. Different dosages [83] and timings [84] have been investigated as well as effects at follow-up [85]. Although demonstrated to be safe [86], ganglioside GM1 did not show established clinical efficacy and was unfortunately set aside [87,88].

## 6. Vascular Dysfunction

Previous studies have shown that at physiological concentration, S1P induces vasculoprotective signaling by stimulating eNOS-derived NO production through engagement of S1P1 and S1P3 [89,90]. In contrast, higher S1P levels promote S1P3-mediated vascular barrier dysfunction [91]. These differences could be explained by the dynamic signaling model proposed by Kerage et al., showing that S1P signaling differentially regulates vascular tone at a dynamic range [92]. This notion is supported by numerous studies that have identified altered sphingolipid metabolism in blood vessels as an important trigger of vascular dysfunction [14,72,93], a key contributor to the pathogenesis of cardiovascular disorders. Ex vivo studies showed impaired endothelial-dependent vasodilation of vessels after ceramide stimulation, an effect that was mediated by increased ROS production and reduced nitric oxide levels NO [94,95]. In their work, Li et al. [96] incubated human endothelial cells with exogenous cell-permeable C6 or C8 ceramide or with bacterial SMase, thus, demonstrating that while ceramides enhance endothelial nitric oxide synthase (eNOS) transcription on the one hand, this causes ROS production to evoke endothelial dysfunction and an impairment in NO-mediated vasorelaxation, on the other. This process appears to be mediated by different mechanisms, such as activation of the NAD(P)H oxidase family—the main enzyme responsible for ROS generation—[97], the interaction with the mitochondrial electron transport chain [98] and the uncoupling of eNOS [96]. This last phenomenon occurs in the presence of a reduced availability of a pivotal enzymatic cofactor required for the synthesis of NO (tetrahydrobiopterin (BH4)) and is caused by an increased oxidation of BH4 itself by ROS [99]. In these conditions, eNOS, which is the primary enzyme responsible for NO generation in the vascular endothelium, contributes to ROS production in lieu of NO. The involvement of sphingolipids in this harmful process is confirmed by the finding that the ceramide-induced reduction of NO is rescued by the treatment with BH4 and by the evidence that the treatment with the eNOS inhibitor L-NG-Nitro arginine methyl ester (L-NAME), limited, in part, endothelial ROS generation [96]. In line with the impairment in NO signaling, further studies have shown that inhibition of ceramide synthesis or lysosomal acid SMase preserves eNOS-mediated signaling and vascular function in diet-induced obesity [13,14] and after TNF-α exposure [93], respectively. S1P also plays an important role in the modulation of vascular reactivity. Chronic S1P-infused wild-type C57BL/6J mice were shown to have increased vasoconstriction and endothelial dysfunction of mesenteric arteries [70], thus, suggesting S1P as an important marker of endothelial dysfunction. These findings were further supported by human observation in which elevation of S1P serum levels was positively correlated with impaired endothelial function as well as with increased vessels contractility [70].

## 7. Diabetes

Diabetes represents a global health issue, with an estimated 415 million people afflicted worldwide, more than 90% of whom had type 2 diabetes (T2DM) [100]. In order to reduce this high burden, one of the most effective ways could be the detection of novel biomarkers able to predict the onset of T2DM. Recent studies have demonstrated that sphingolipids, in particular ceramides, could contribute to insulin resistance by inhibiting the activation of protein kinase B (PKB), which leads to the blocking of the translocation of glucose transporter 4 (GLUT4) and the consequent suppression of insulin-stimulated glucose uptake and glycogen synthesis. Incubation of muscle cells with C2 ceramid stimulates an atypical member of PKC family, PKCζ, which phosphorylates the PKB’s PH domain on Thr34. In the absence of insulin, PKB is present in the cytosol in its inactive state where it forms a pre-activation complex with 3-phosphoinositide-dependent protein kinase-1 (PDK1) [101]. After insulin stimulation, this complex translocates to the plasma membrane and hooks onto the membrane lipid phosphatidylinositol-3,4,5-trisphosphate (PIP3) [102], where it is converted to the active phosphorylated form, p-PKB [101,103], and causes insulin-induced glucose uptake into cells by stimulating the translocation of GLUT4 to the cell surface [104]. The phosphorylation induced by C2 ceramide reduces the effects induced by insulin on the glucose metabolism through the inhibition of the binding of PIP3 to the PH domain. Mechanistically, in the presence of ceramide, the hormone is not only enabled to induce PKB’s activation, but loses its ability to dissociate the PKCζ/PKB complex [105]. Several studies have also demonstrated that C2 ceramide is able to completely inhibit insulin-stimulated PKB phosphorylation at both the T308 and S473 regulatory sites, causing a 90–95% loss in constitutive kinase activity [106]. The dephosphorylation induced by ceramide is entirely abolished by pretreatment with okadaic acid, a phosphatase inhibitor [107]. This discovery suggests that C2 ceramide is implied in the development of insulin resistance by keeping PKB in an inactive state as well as, probably, through the involvement of the principal phosphatase that dephosphorylates the serine/threonine protein kinase in adipocytes [108]: protein phosphatase 2A (PP2A), which is okadaic-acid-sensitive [107]; this hypothesis was confirmed by the suppression of the sphingolipid’s effects through the expression of a PP2A inhibitor: the SV40 small T antigen [103]. In the presence of C2 dihydroceramide, a biologically inactive ceramide analog, these effects were abolished [107]. Insulin resistance could also be caused by the blocking of the insulin-stimulated phosphorylation of a hormone receptor substrate IRS-1 induced by ceramides [109,110,111] and by the ceramide-stimulated phosphorylation of its inhibitory serine residue [112], resulting in a severe impairment in insulin signal transduction. Indeed, ceramides have been shown to be able to regulate mTORC1 activity [113] and to activate some kinases, such as c-Jun N-terminal kinase (JNK) and IκKβ [114], involved in inactivation due to the phosphorylation of serine 307 of IRS-1 [113,115,116]. Several studies have shown that SphK2, the enzyme predominantly responsible for catalyzing the conversion of dihydrosphingosine (dhSph) to dihydro-S1P (dhS1P), is involved in both core features of diabetes: pancreatic β-cell lipotoxicity and dysfunction [117], and insulin resistance and glucose intolerance [118]. It is known that excessive ectopic deposition of sphingolipids in the pancreas precedes organ failure, β-cell dysfunction and death [119] and the consequent reduced insulin production [120], a phenomenon known as gluco-lipotoxicity [121]. Indeed, due to their capacity to increase the activity of phosphatase PP2A, ceramides are able to catalyze, in the islet of Langerhans and pancreatic β-cell lines, the inactivation of the extracellular-signal-regulated kinases (ERKs), leading to a decrease in proinsulin gene transcription [122]. Moreover, ceramides inhibit the nuclear translocation of PDX-1 and Mafa, two important transcription factors for insulin-induced gene expression [119], and may block the glucose-induced expression of PASK (Per-Arnt-Sim domain-containing kinase), a serine/threonine protein kinase involved in the control of pancreatic islet hormone release and insulin sensitivity [123]. In addition, pancreatic sphingolipid accumulation leads to endoplasmic reticulum (ER) stress [124], mitochondrial dysfunction [125] and NADPH oxidase activation and consequent ROS production [126], which, in turn, induce β-cell apoptosis [121], as well as through the stimulation of pro-apoptotic agents, such as caspase [127], serine/threonine protein phosphatase 1 (PP1) [114] and the SAPK/JNK signaling pathway [128]. At last, ceramides appear to play a role in both β-cell apoptosis [129] and insulin resistance [130] induced by proinflammatory cytokines, such as tumor necrosis factor-alpha (TNF-α), interleukin-1 beta (IL-1β) and interferon-gamma (IFN-γ) [131]. 

Multivariable analysis demonstrated that a small number of specific sphingolipid species were independent predictors of future cardiovascular events and deaths in individuals with T2DM [132]. In a population-based cohort study of approximately 2000 Chinese individuals, the authors identified a novel panel of sphingolipids positively associated with increased T2DM risk [133]. These findings are consistent with other previous studies, in which higher concentrations of distinct SM and ceramides have been correlated with incident T2DM [134,135,136]. In their work, Chen and collaborators [136] highlighted the involvement of sphingolipids in the onset of diabetes. They demonstrated that dihydrosphingosine (dhSph) and dihydro-S1P (dhS1P) could be candidates as potential biomarkers for the early diagnosis of T2DM. They showed that, in patients who developed diabetes, the levels of dhS1P and its ratio to dhSph were elevated 4.2 years before the disease was diagnosed [136]. Similarly, quantitative analysis of circulating sphingolipids in individuals from two prospective cohorts revealed a significant increase in specific long-chain fatty-acid-containing dihydroceramides in humans up to 9 years before T2DM onset [134].

A growing number of studies suggest that one atypical sphingolipid (deoxyceramide-Cer(m)) species, which is formed by abnormal SPT activity, accumulates in patients with metabolic syndrome or T2DM [137,138,139]. In detail, variants in the genes SPTLC1 or SPTLC2 induce a shift in the substrate preference of SPT from serine to alanine and glycine, thereby leading to the formation of the two atypical deoxy-sphingoid bases: 1-deoxy-sphinganine and 1-deoxymethyl-sphinganine, defined as 1-deoxysphingolipids (1-DSL) [140]. Both of these metabolites lack the C1 hydroxyl group of sphinganine and, therefore, cannot be converted to more complex sphingolipids or degraded [140]. Deoxysphingolipids have been shown to cause toxicity in various cell types, including neurons, myoblasts, β cells and retinal cells [141,142,143,144,145]. In vitro studies reported that 1-DSL significantly lowered the viability of C2C12 myoblasts in a concentration- and time-dependent manner and induced necrosis, apoptosis as well as cellular autophagy [145]. Additionally, 1-DSL significantly compromises the functionality of skeletal muscle cells, by impairing migration, differentiation and insulin-stimulated glucose uptake. The authors suggested that increased levels of 1-DSL detected in T2DM patients may contribute to the pathophysiology of muscle dysfunction associated with this disease [145].

## 8. Obesity

Growing evidence indicates that ceramides can affect different metabolic pathways depending on the specific fatty acyl chain lengths, a process regulated by six ceramide synthases, namely (CerS1–CerS6), whose expression differs throughout the body [146]. In line with this discovery, the reduction in specific ceramide pools in a tissue-specific manner resulted in a significant improvement in metabolic phenotype in a mouse model of obesity [147].

Genetic mouse studies suggest that the specific sphingolipid C16:0 ceramide produced by CerS6 represents a critical player for the onset of insulin resistance. Using an antisense oligonucleotide (ASO) approach in an obese diabetic ob/ob mouse model, CerS6 knockdown significantly reduced body-weight gain and whole-body fat and fed/fasted blood-glucose levels, accompanied by a 1% reduction in glycated hemoglobin [148]. Moreover, ASO-treated mice exhibited improved insulin sensitivity and oral glucose tolerance [148]. Accordingly, a critical role of CerS6-dependent C16:0 ceramide production emerged in the regulation of adipose tissue function in obesity conditions. CerS6 mRNA expression and C16:0 ceramide levels were found elevated in the white adipose tissue (WAT) of 439 obese subjects, and this increase was correlated with insulin resistance, body-fat content and hyperglycemia [149]. Accordingly, deficiency in CerS6 in mice reduced ceramide concentration and prevented high-fat-diet-mediated obesity and glucose intolerance [149]. Of note, these effects were also replicated in brown adipose tissue- and liver-specific CerS6 knockout mice [149]. A central role of CerS1-derived C18:0 ceramide was demonstrated in the development of obesity-associated insulin resistance. Skeletal muscle is considered a major tissue involved in the regulation of glucose and lipid metabolism. Global or skeletal muscle-specific deletion of CerS1 resulted in reduced C18:0 ceramide content, improved insulin-stimulated suppression of hepatic glucose production and systemic glucose homeostasis in a Fibroblast growth factor 21 (FGF21)-dependent manner [150,151]. In line with this, cell-specific deficiency in CerS5- and CerS6- derived C16:0 failed to prevent insulin resistance in obese mice [150]. Genome-wide association studies identified the SPT suppressor ORMDL3 as an obesity-related gene, and its expression in human subcutaneous WAT was inversely correlated with BMI [152]. Importantly, ORMDL3 is downregulated in the WAT of obese mice and humans, and its deletion resembles the metabolic phenotypes of obesity induced by a high-fat diet (HFD) [153]. In fact, Ormdl3 deficiency in mice raised WAT ceramide levels, body weight and insulin resistance, all prevented by an inhibition in ceramide production with the SPT inhibitor myriocin [153]. A diverse range of clinical trials have investigated the relationship between obesity and sphingolipids. Muscle sphingolipids during exercise training evaluated with VO2 max, in three different subgroups of patients (athletes, obese and DM type II), showed a significant direct relation to levels of muscle C18:0 ceramide and an inverse relation with insulin resistance [154,155]. In the same subgroups of patients, a study on intracellular localization thanks to muscle biopsy showed that C18:0 ceramide is inversely related to insulin sensitivity [156]. Interestingly, after exercise training, improvements in insulin sensitivity were associated with a significant improvement in BMI, adiposity and a reduction in C14:0, C16:0, C18:0 and C24:0 ceramide, in both obese and T2DM patients [157]. Patients with metabolic syndrome in treatment with pioglitazone for 6 months showed a significant reduction in C16:0, C18:0, C20:0, C22:0 and C24:1 as compared to placebo. However, correlation with insulin sensitivity was not homogeneously positive [158]. Studies on human subjects demonstrated high serum levels of SM species with distinct saturated acyl chains (C18:0, C20:0, C22:0 and C24:0) in the obese as compared to age-matched healthy subjects [159]. Notably, the authors found that these sphingolipids were closely associated with the parameters of obesity, insulin resistance, lipid metabolism and liver function, thus, their involvement in the development of metabolic syndrome and their use as novel biomarkers of metabolic syndrome [159]. Similarly, dihydroceramide species 18:0, 20:0, 22:0 and 24:1 as well as sphingomyelin species 31:1 and 41:1 were significantly correlated with waist circumferences, a clinical marker of central obesity [160].

Even though each had a different scope, these studies highlighted the relation between sphingolipids and obesity. Evidence and future clinical trials and larger-scale human metabolomics studies are still needed.

## 9. Conclusions and Future Perspectives

Cardiovascular diseases represent a major burden for global health, reducing quality of life and increasing mortality. All the evidence summarized in this review sheds some light on the mechanisms by which altered sphingolipid metabolism is involved in the pathogenesis of CVDs, paving the way for the use of these sphingolipids as promising biomarkers to improve risk stratification for cardiovascular and cerebrovascular diseases. The overall outcomes from clinical studies are summarized in Table 1. Cumulatively, these studies suggest that monitoring the imbalance in sphingolipid levels in the circulation could offer relevant tools to better assess the progression and/or severity of metabolic, cardio- and cerebrovascular disease. Therefore, using multiple sphingolipids as biomarkers in combination with clinical parameters might be useful to guide interventions focused on the prevention of adverse cardiovascular risk factors that might improve patients’ survival. Moreover, in light of recent findings, sphingolipids hold great promise for improvements in treatment strategies. Some approaches may include the development of enzyme inhibitors, such as sphingosine kinases, sphingomyelinases or ceramidases, that catalyze ceramide catabolism or its conversion to the bioactive form of S1P. Despite the efficacy of common pharmacological treatments, a large percentage of patients remain non-responders to drug therapy. Therefore, targeting sphingolipid metabolism appears to be important for the development of future therapeutic agents or to increase current treatment effectiveness.

## Figures and Tables

**Figure 1 biomolecules-13-00168-f001:**
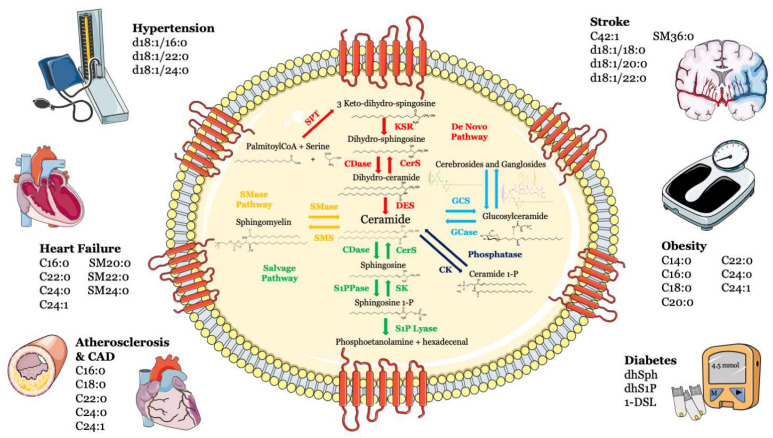
Metabolism and structure of sphingolipids and their implication in cardio- and cerebrovascular diseases. Sphingolipid metabolism and structure are illustrated inside the cell. Ceramide is the heart of the sphingolipid metabolic pathway. Ceramide can be synthesized through several steps: (i) de novo synthesis pathway starting from l-serine and palmitoyl-CoA (red); (ii) SMase pathway, through hydrolysis of sphingomyelin (yellow); (iii) salvage pathway, long-chain sphingoid bases are reused to form ceramide through the action of ceramide synthase (green); (iv) or through hydrolysis of glycosphingolipids and sulfatites (azure). Ceramide can also be synthesized from ceramide-1-phosphate through the action of ceramide-1-phosphate phosphatase (blue). The main cardiovascular diseases and risk factors in which sphingolipids may be used as biomarkers are summarized outside the cell. Abbreviations: serine palmitoyl-CoA-acyltransferase (SPT), 3-ketosphinganine reductase (KSR), (dihydro)-ceramide synthase (CerS), ceramide desaturase (DES), ceramide kinase (CK), glucosylceramide synthase (GCS), glucosyl ceramidase (GCase), ceramidase (CDase), sphingosine-1-phosphate lyase (S1P lyase), sphingosine kinase (SK), sphingosine 1-phosphate phosphatase (S1PPase), sphingomyelin (SM) synthase (SMS), sphingomyelinase (SMase).

**Table 1 biomolecules-13-00168-t001:** Summary of the main findings from the clinical studies discussed in the review.

Clinical Condition	Study	Model	Major Findings
Atherosclerosis and CAD	Jiang et al. [48]	Plasma from 279 biethnic patients with coronary artery disease (CAD) was related with 277 controls.	Higher plasma sphingomyelin (SM) level in CAD patients than in control subjects; higher ratio of SM to SM + phosphatidylcholine (PC) in patients than in controls.
	Gao et al. [49]	Plasma from 100 patients with Coronary atherosclerosis and 100 controls was compared.	Increased levels of 24 metabolites and decreased of 18 metabolites in early CAS patients compared with the controls; nine metabolites resulted useful as biomarkers to distinguish early-stage CAS patients from controls.
	Hilvo et al. [50]	Plasma of non-diabetic patients with established coronary heart disease (CHD), stratified in 25 patients receiving RG7652 (PCSK9 inhibitor) and 15 placebo, was analyzed.	PCSK9 inhibition decrease plasma levels of several lipid classes, including sphingolipids (dihydroceramides, glucosylceramides, sphingomyelins, ceramides), cholesteryl esters and free cholesterol.
	Croyal et al. [51]	Plasma from 102 patients with T2DM before and after fenofibrate treatment (200 mg/day) was correlated.	Ceramide levels decreased in 73.5% of patients; reduction of plasma apoC-II, apoC-III, apoB100, SMase, with an increase in apoA-II and adiponectin levels.
	Poss et al. [52]	Correlation between serum of 462 individuals with familial coronary artery disease (CAD) and 212 population-based controls.	30 sphingolipids were found elevated in serum of patients with CAD compared with healthy controls.
	Tu et al. [53]	Plasma from 553 patients with definite or suspected CAD.	High ratio of Cer (d18:1/24:1) to Cer (d18:1/24:0), female gender, HbA1c%, unstable angina (UAP) and acute myocardial infarction (AMI) diagnosis during hospitalization were related with severe coronary artery stenosis.
	Mantovani et al. [54]	Plasma from 167 patients with established or suspected CAD, who underwent urgent or elective coronary angiography.	Higher levels of plasma Cer(d18:1/20:0), Cer(d18:1/22:0) and Cer(d18:1/24:0) were associated with the presence of LAD stenosis ≥ 50%.
	Hilvo et al. [55]	Plasma from 3789 individuals of the WECAC (The Western Norway Coronary Angiography Cohort), 5991 individuals of the LIPID (Long-Term Intervention with Pravastatin in Ischaemic Disease) trial, and 1023 individuals of the KAROLA (Langzeiterfolge der KARdiOLogischen Anschlussheilbehandlung)	Higher ceramides levels were found in subjects with CAD.
	Laaksonen et al. [8]	Plasma from 160 individuals of the Corogene cohort study including stable CAD patients; 1637 individuals of the prospective Special Program University Medicine—Inflammation in Acute Coronary Syndromes (SPUM-ACS) cohort.	Ceramides were associated with CV death in all studies, independent of other lipid markers and C-reactive protein.
	Wang et al. [56]	Plasma from 980 participants from the PREDIMED trial (Prevención con Dieta Mediterránea); 230 incident cases of CVD and 787 randomly selected participants at baseline (including 37 overlapping cases).	The ceramide score, calculated as a weighted sum of concentrations of four ceramides, was correlated with a 2.18-fold higher risk of CVD.
	Meeusen et al. [57]	Plasma from 495 participants before nonurgent coronary angiography.	Coronary artery disease was identified in 265 (54%) cases; the hazard ratios were 1.50 (1.16–1.93) for Cer(16:0), 1.42 (1.11–1.83) for Cer(18:0), 1.43 (1.08–1.89) for Cer(24:1), and 1.58 (1.22–2.04) for the ceramide risk score.
	Peterson et al. [58]	Plasma from 2642 participants of the Framingham Heart Study (FHS) and 3134 participants of the Study of Health in Pomerania (SHIP).	During a mean follow-up of 6 years in FHS, there were 88 coronary heart disease (CHD) and 90 heart failure (HF) events and 239 deaths. During a median follow-up time of 5.75 years for CHD and HF and 8.24 years for mortality, in SHIP there were 209 CHD and 146 HF events and 377 deaths. C24:0/C16:0 ceramide ratios were inversely associated with incident CHD and inversely associated with incident HF; C24:0/C16:0 and C22:0/C16:0 ceramide ratios were inversely associated with all-cause mortality.
	Pan et al. [59]	Plasma from 304 patients and 52 healthy individuals divided in four groups: 52 control group, 98b stable angina pectoris (SAP) group, 92 unstable angina pectoris (UAP) group, 114 and acute myocardial infarction (AMI) group.	Higher levels of sphingomyelin (SPM) in patients with UAP and AMI compared with the controls and SAP participants. Higher ceramide levels and S-SMase activity in patients with UAP and AMI than controls and SAP participants.
Heart failure	Ji et al. [12]	Serum and myocardial tissue from 65 patients with advanced heart failure (HF).	Increase of long–chain ceramides in myocardium and serum of patients with advanced HF. Blocking the serine palmitoyl transferase (SPT), the enzyme of the pathway of ceramide synthesis, higher ceramides were found.
	Anroedh et al. [16]	Blood from 581 patients who underwent diagnostic coronary angiography or percutaneous coronary intervention for stable angina pectoris (SAP) or acute coronary syndrome (ACS).	During a median follow-up of 4.7 years, 155 patients (27%) had MACEs. Cer(d18:1/16:0) concentration was related with MACEs; concentrations of Cer(d18:1/16:0), Cer(d18:1/20:0), Cer(d18:1/24:1), and their ratios to Cer(d18:1/24:0) were associated with the composite endpoint death or nonfatal ACS.
	Perez-Carrillo et al. [65]	Myocardial tissue samples from the left ventricle of 52 subjects: 42 patients with HF (non-ischemic dilated and ischemic cardiomyopathy patients) undergoing cardiac transplantation and 10 samples from non-diseased donor hearts.	Sphingolipid metabolism gene dysregulation was found in HF human cardiac tissue, with the major changes occurring in the expression of genes involved in the de novo and salvage pathways; S1P is enhanced in HF cardiac tissue.
	Lemaitre et al. [66]	Plasma from 4249 patients.	Identified 1179 cases of incident heart failure among 4249 study participants. With Cox regression, higher levels of Cer-16 and SM-16 were correlated with higher risk of incident heart failure. In contrast, higher levels of Cer-22, SM-20, SM-22 and SM-24 were correlated with lower risk of heart failure.
Hypertension	Spijkers et al. [7]	Plasma from 12 patients with stage 1 hypertension; 19 patients with stage 2 and 3 hypertension and 18 normotensive controls.	Higher ceramide levels in patients with stage 2 or 3 hypertension compared to healthy normotensive controls.
	Jujic et al. [71]	Plasma from 1046 individuals of the Malmö Offspring Study (MOS).	Individuals with systolic BP ≥140 mm Hg had higher S1P plasma concentrations compared with subjects with BP <120 mm Hg independent of age and sex.
	Di Pietro et al. [72]	Plasma from the Campania Salute Network Registry divided in 71 patients with hypertension, 36 healthy donor control subjects and 36 hypertensive patients with no diagnosis of PAD. Plasma from 270 individuals of the Moli-Sani study stratified into normotensive subjects (n = 89) and controlled (n = 91) or uncontrolled hypertensive subjects (n = 90).	Increased plasma ASMase activity and levels of S1P, in hypertensive subjects; the increase was more pronounced in hypertensive subjects with uncontrolled blood pressure.
	Yin et al. [73]	Plasma from 920 patients with hypertension.	During mean 2.3-year follow-up, 71 patients had MACE. Cer(d18:1/16:0), Cer(d18:1/22:0), and Cer(d18:1/24:0) were highly significant in predicting MACE.
Stroke	Sheth et al. [78]	Plasma from 14 patients with acute stroke symptoms.	Higher sphingolipid scores in patients with true stroke compared to patients who hsd non-stroke causes of their symptoms.
	Lee et al. [79]	Plasma from 87 patients with acute ischemic stroke (AIS) and 30 nonstroke controls.	Decreased S1P and very-long-chain ceramides in AIS patients compared to non-stroke controls; increased long-chain ceramides in AIS patients.
	Lee et al. [80]	Plasma from 75 AIS patients who underwent endovascular thrombectomy before (T1), immediately after (T2), and 24 h after (T3) the procedures and 19 controls.	Higher plasma levels of long-chain ceramides Cer (d18:1/16:0) at all three time points, Cer (d18:1/18:0) at T1 and T3, and Cer (d18:1/20:0) at T1 and very-long-chain ceramide Cer (d18:1/24:1) at T1 in AIS patients compared to controls. Lower plasma levels of sphingosine-1-phosphate in AIS patients compared to controls at all three time points.
	You et al. [81]	Plasma from 20 patients with large artery atherosclerosis (LAA), 20 patients with age-related cerebral small vessel disease (CSVD), 10 patients with Fabry disease and 14 controls.	Cer (d36:3), Cer (d34:2), Cer (d38:6), Cer (d36:4) and Cer (d16:0/18:1) were increased in LAA; SM (d34:1), Cer (d34:2), Cer (d36:4), Cer (d16:0/18:1), Cer (d38:6), Cer (d36:3) and Cer (d32:0) were increased in age-related CSVD. Cer (d36:4) and SM (d34:1) were increased in age-related CSVD compared with LAA. Total trihexosyl ceramides were higher in Fabry group compared with control; SM (d34:1) was increased in Fabry group.
Vascular dysfunction	Siedlinski et al. [70]	Serum from 82 patients.	Higher S1P serum levels were correlated with impaired endothelial function as well as with increased vessels contractility.
Diabetes	Alshehry et al. [132]	Plasma from 3779 patients with T2DM.	Plasma lipids species sphingolipids, phospholipids (including lyso- and ether- species), cholesteryl esters, and glycerolipids were related to future cardiovascular events and cardiovascular death.
	Yun et al. [133]	Plasma from 1974 individuals followed-up for 6 years T2DM.	11 novel and 3 reported sphingolipids, namely ceramides (d18:1/18:1, d18:1/20:0, d18:1/20:1, d18:1/22:1), saturated sphingomyelins (C34:0, C36:0, C38:0, C40:0), unsaturated sphingomyelins (C34:1, C36:1, C42:3), hydroxyl-sphingomyelins (C34:1, C38:3), and a hexosylceramide (d18:1/20:1), were associated with incident T2DM.
	Wigger et al. [134]	Plasma from 250 individuals of two different prospective cohorts who developed T2DM.	Specific long-chain fatty-acid-containing dihydroceramides were higher in the plasma of individuals up to 9 years before disease onset.
	Chew et al. [135]	Plasma from plasma of 2302 ethnically-Chinese Singaporeans.	Two distinct sphingomyelins, d16:1/C18:0 and d18:1/C18:0 were related to a higher risk of T2DM.
	Chen et al. [136]	Serum from a total of 2486 non-diabetic adults at baseline, 100 subjects who developed T2DM after a mean follow-up of 4.2-years and 100 control subjects matched strictly with age, sex, BMI and fasting glucose.	Compared to the control group, medians of serum dhS1P and dhS1P/dhSph ratio at baseline were elevated prior to the onset of T2DM. Each SD increment of dhS1P and dhS1P/dhSph ratio was related to 53.5% and 54.1% higher risk of incident diabetes, respectively.
	Fridman et al. [137]	Plasma from 75 individuals stratified in: 19 lean controls (LC), 19 with obesity, 18 with obesity and T2DM without diabetic neuropathy (DN) (ob/T2DM), and 19 with obesity, T2DM and DN (Ob/T2DM/DN).	Increased 1-deoxydihydroceramides across these four groups. 1-deoxydihydroceramide species were higher in ob/T2DM/DN versus LC group. No significant differences in 1-deoxydihydroceramides were found between the ob/T2DM and ob/T2DM/DN groups.
	Mwinyi et al. [138]	Plasma from 605 non-diabetic individuals.	1-deoxysphingolipids were higher in individuals who developed T2DM during 5-year follow-up; 1-deoxy-sphinganine and 1-deoxy-sphingosine were predictive for T2DM.
	Othman et al. [139]	Plasma from 339 individuals.	Increased 1-deoxysphingolipids levels in patients with metabolic syndrome, impaired fasting glucose, and T2DM. Patients who developed T2DM during 8-year follow-up period showed an increase in 1-deoxysphingolipids levels at baseline compared with those who did not develop T2DM until the end of the study.
Obesity	Turpin et al. [149]	White adipose tissue (WAT) from 439 obese subjects.	CERS6 mRNA expression and C16:0 ceramides were elevated in adipose tissue of obese humans, and increased CERS6 expression correlates with insulin resistance.
	Pan et al. [152]	Subcutaneous abdominal adipose needle biopsy from 10,197 individuals.	Identified the SPT suppressor ORMDL3 as an obesity-related gene, and its expression in human subcutaneous WAT was inversely correlated with BMI.
	Bergman et al. [154]	Serum and percutaneous needle biopsy between the greater trochanter of the femur and the patella was taken from 14 obese sedentary individuals, 15 type 2 diabetic patients and 15 endurance trained athletes.	Muscle C18:0 ceramide, dihydroceramide and glucosylceramide species were inversely associated with insulin sensitivity without differences in total ceramide, dihydroceramide, and glucosylceramide concentration. Muscle C18:0 dihydroceramide was correlated to markers of muscle inflammation.
	Bergman et al. [155]	Serum from 14 obese sedentary controls, 15 individuals with T2DM, and 15 endurance-trained athletes.	Basal C18:0, C20:0, and C24:1 ceramide and C18:0 and total dihydroceramide were higher in T2DM and, along with C16:0 ceramide and C18:0 sphingomyelin, correlated positively with insulin resistance.
	Perreault et al. [156]	Muscle biopsies from 15 lean individuals, 16 endurance-trained athletes, and 12 obese men and women with T2DM and 15 without T2DM.	Sarcolemmal sphingomyelins were inversely correlated to insulin sensitivity, with the strongest relationships found for the C18:1, C18:0, and C18:2 species.
	Kasumov et al. [157]	Plasma from 24 adults with obesity and normal glucose tolerance (NGT, n = 14) or diabetes (n = 10).	Plasma ceramides were similar for the subjects with obesity and NGT and the subjects with diabetes. Exercise reduced body weight and adiposity and increased peripheral insulin sensitivity in both groups. Reduced plasma levels of C14:0, C16:0, C18:1, and C24:0 ceramide in all subjects following the intervention.
	Warshauer et al. [158]	Plasma from 37 subjects with metabolic syndrome receiving pioglitazone or placebo.	Decreased plasma levels of C18:0, C20:0, C24:1, dihydroceramide C18:0, dihydroceramide C24:1, lactosylceramide C16:0 and the hexosylceramides C16:0, C18:0, C22:0 and C24:1 in patients in treatment with pioglitazone compared to placebo.
	Hanamatsu et al. [159]	Serum from 12 obese participants and 11 controls.	C18:0 and C24:0 levels were significantly higher in the obese individuals. C20:0 and C22:0 levels tended to be higher in the obese group than in the control group. SM C18:0, C20:0, C22:0 and C24:0 correlated with the parameters for obesity, insulin resistance, liver function and lipid metabolism, respectively.
	Mamtani et al. [160]	Plasma of 1208 Mexican Americans from 42 extended families.	Dihydroceramide species 18:0, 20:0, 22:0, and 24:1 were genetically associated with Waist circumference (WC). Two sphingomyelin species (31:1 and 41:1) were also related with WC.

## Data Availability

Data sharing not applicable.

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
