# Peer review of "The Dark Side of Sphingolipids: Searching for Potential Cardiovascular Biomarkers"

_biomolecules, 2023, doi:10.3390/biom13010168_

Round 1

Reviewer 1 Report

The review manuscript by Di Pietro nicely summarize the current knowledge of sphingolipids in cardiovascular diseases. The authors nicely describe the current knowledge and the limitations of the use of sphingolipids as biomarkers for several cardiovascular conditions as well as their known and plausible modes of action. Overall is a very nicely written review. I only miss one point that to my point of view is compulsory, there is a need to nicely illustrate the global role of sphingolipids in cardiovascular diseases by providing a summarizing figure.

Minor comments

Please revise sentence on page 3, lanes 119-120, since it seems to be incoherent. 

Author Response

Reviewer #1

The review manuscript by Di Pietro nicely summarizes the current knowledge of sphingolipids in cardiovascular diseases. The authors nicely describe the current knowledge and the limitations of the use of sphingolipids as biomarkers for several cardiovascular conditions as well as their known and plausible modes of action. Overall is a very nicely written review.

I only miss one point that to my point of view is compulsory, there is a need to nicely illustrate the global role of sphingolipids in cardiovascular diseases by providing a summarizing figure.

Minor comments

Please revise sentence on page 3, lanes 119-120, since it seems to be incoherent.

We thank the Reviewer for his/her comments and valuable suggestions. Accordingly, we have added one summarizing figure, which, we hope, would help to visualize the major findings of our manuscript, page 2, figure 1. In addition, the sentence on page 3 has been revised to clearly convey the message.

Reviewer 2 Report

The review is well written and comprehensive. 

The following changes could make the review an easy read.

a) Both the abstract and the conclusion are a bit too vague. It will be great to see specific suggestions regarding the potential of sphingolipids to be used as biomarkers or proteins from the sphingolipid synthesis pathway that could be used as drug targets. Some detailed discussions about these aspects in the abstract as well as in the conclusions will make the article an informative read.

2) It will be great to have a figure showing the overview of sphingolipid structure and metabolism , highlighting the role of sphingolipids and their clinical implications. 

3) The authors should consider presenting the different findings in each of the clinical conditions in a tabular form.

Author Response

Reviewer #2

The review is well written and comprehensive.

The following changes could make the review an easy read.

  1. a) Both the abstract and the conclusion are a bit too vague. It will be great to see specific suggestions regarding the potential of sphingolipids to be used as biomarkers or proteins from the sphingolipid synthesis pathway that could be used as drug targets. Some detailed discussions about these aspects in the abstract as well as in the conclusions will make the article an informative read.

We thank the Reviewer for his/her careful review, which helped us to improve the manuscript. Accordingly, these aspects have been added in the revised version of the manuscript. In particular, we have completely revised the last paragraph by adding insights on future perspectives regarding the use of sphingolipids as biomarkers and/or drug targets: page 1; lines 26-32; page 10, lines 454-468.

2) It will be great to have a figure showing the overview of sphingolipid structure and metabolism, highlighting the role of sphingolipids and their clinical implications.

According to the Reviewer’s suggestion, we have added a summarizing figure, which, we hope, would help to visualize the major findings of our manuscript: page 2, figure 1.

3) The authors should consider presenting the different findings in each of the clinical conditions in a tabular form.

We very thank the Reviewer for his/her suggestion. Accordingly, in the revised version of the manuscript we have included a table summarizing the main findings in each of the clinical conditions covered in our manuscript: pages, 11-21.

Reviewer 3 Report

The authors reviewed the detrimental effect of Sphingolipids. Even though all of the contents are well-summarized, there are no comments of what we should do in future.

Indeed, this persepctive is not appropriate structure for review article. The authors should add this insight in this manuscript.

Author Response

Reviewer #3

The authors reviewed the detrimental effect of Sphingolipids. Even though all of the contents are well-summarized, there are no comments of what we should do in future.

Indeed, this perspective is not appropriate structure for review article. The authors should add this insight in this manuscript.

We thank the Reviewer for his/her comments and criticisms. Accordingly, we have revised the last paragraph by adding future perspectives regarding the use of sphingolipids as biomarkers and/or potential drug targets: page 1; lines 26-32; page 10, lines 454-468.

Round 2

Reviewer 2 Report

The authors have made extensive changes to the review, and could be accepted .